# Defining destigmatizing design guidelines for use in sexual health-related digital technologies: A Delphi study

**Abdul-Fatawu Abdulai** [1]*, **A. Fuchsia Howard**[1], **Paul J. Yong**[2], **Leanne M. Currie**[1]

1 School of Nursing, University of British Columbia, T201-2211 Westbrook Mall, Vancouver, BC, Canada,
2 Department of Obstetrics and Gynaecology, University of British Columbia, Vancouver, BC, Canada

* Fatawu.Abdulai@ubc.ca

**Data Availability Statement:** Data for this study are available upon a reasonable request via the University of British Columbia Office of Research Ethics by e-mail at RSIL@ors.ubc.ca.

## Abstract

### Objective

Stigma has been recognized as a significant issue in sexual health, yet no specific guidelines exist to support digital health development teams in creating stigma-alleviating sexual health digital platforms. The purpose of this study was to develop a set of design guidelines that would serve as a reference point for addressing stigma during the design of sexual health-related digital platforms.

### Materials and Methods

We conducted a 3-round Delphi study among 14 researchers in stigma and sexual health. A preliminary list of 28 design guidelines was generated from a literature review. Participants appraised and critiqued the clarity and usefulness of the preliminary list and provided comments for each item and for the overall group of items at each round. At each round, a content validity index and an interquartile range were calculated to determine the level of consensus regarding the clarity and usefulness of each guideline. Items were retained if there was high consensus or were dropped if there was no consensus after the three rounds.

### Results

Nineteen design guidelines achieved consensus. Most of them were content-related guidelines and sought to address the emotional concerns of patients that could potentially aggravate stigma. The findings also reflected modern stigma management strategies of making stigma a societal attribute by challenging, exposing, and normalizing stigma attributes via web platforms.

### Conclusion

To address stigma via digital platforms, developers should not just concentrate on technical solutions but seriously consider content-related and emotional design components that are likely to result in stigma.

**Funding:** This study was funded by the UBC School of Nursing Lyle Creelman Endowment Fund and the Designing for People NSERC-CREATE program. The funders had no role in study design, data collection and analysis, decision to publish, or preparation of the manuscript.

**Competing interests:** The authors have declared that no competing interests exist.

# Introduction

Sexual health-related conditions, herein described as a group of health issues or clinical syndromes that affects an individual's sexuality or sexual functioning, have attracted widespread attention among healthcare providers and policymakers across the globe [1]. Stigma is a well-documented barrier to engagement in care, medication adherence, and health-seeking activities among people living with a variety of sexual health-related conditions [2]. Stigma manifests in two main forms: internalized stigma and public stigma [3]. Internalized stigma occurs when a person perceives their situation as a negative attribute with resulting decreases in self-esteem and self-worth, while public stigma manifests as overt discriminatory practices targeted at a group with attributes that are perceived to be negative [3]. Individuals encountering stigma report several challenges including poor decision-making, poor interaction with family and friends with its associated consequences of social withdrawal, disclosure-associated anxiety, and reduced sexual well-being [4,5]. The stigma faced by people with sexual health-related conditions is more pronounced in conventional health settings such as in-person visits at sexually transmitted infection (STI) clinics than in digital interventions [6]. For example, showing up at a sexual health clinic could reveal that a person has a sexual health condition, and having the sexual health condition may be considered a negative attribute which may contribute to both internal and public stigma. Despite global efforts in combating sexual health-related conditions, the stigma associated with such health issues continues to hamper progress [7]. Even though many biomedical and structural interventions have been developed for the management of sexual health-related conditions, the stigma associated with in-person visits and face-to-face interactions often poses a challenge to the effective utilization of these services [8].

Given the stigma associated with in-person clinical encounters and interpersonal interactions, various digital health interventions have been developed to complement, and in some cases replace, conventional health services [9–14]. Digital health interventions may be more useful for addressing sexual health-related stigma because people are more likely to use technology-based interventions for health problems that are perceived as embarrassing, stigmatizing, and difficult to discuss face-to-face [15]. However, the way in which digital health technologies are designed or deployed could inadvertently reinforce, (re)produce, or perpetuate stigma among users who may find the technology content or functions to be emotionally distressing [10].

Previous studies have revealed how some digital health technologies not only fail to address the stigma of sexual health issues but may also contain interface components that could inadvertently aggravate stigma among end-users [13,16]. For instance, people living with sexual health-related conditions can link the appearance of sexual health-related content to an existing public stigma, thereby resulting in a digital health technology that elicits stigma. The inability of digital health technologies to alleviate stigma or the possibility of provoking stigma via digital platforms suggests a possible lack of a structured set of design guidelines to help address stigmatizing components during the creation of such digital platforms. The absence of structured destigmatizing design guidelines may make stigma a lesser consideration in digital health design despite being an outcome of interest for healthcare providers, patients, and web users alike. In an early project, we started to develop destigmatizing design guidelines but recognized that the field would benefit from a rigorous expert review [17]. Thus, the purpose of this study was to develop a set of destigmatizing design guidelines that would serve as a reference point for addressing stigma during the design of sexual health-related digital platforms. Design guidelines are generally described as "rules of thumb" or "universally applicable laws, rules, or considerations" that are used as a foundation for a pleasant design [18]. The guidelines produced from this study might help digital health development teams (e.g., software

designers and content creators) to determine which digital health content or features could be perceived to be stigmatizing by end-users [19]. We also expect the findings to help move the emphasis of digital health development from largely employing technical solutions to considering other non-technical factors that might alleviate or aggravate stigma.

## Materials and methods

This study adopted a Delphi technique to develop a set of destigmatizing design guidelines. A Delphi study was chosen because it supports the development of items through consensus-building among a group of experts on a given topic. A panel of experts in stigma and sexual health research were consulted via an online survey and asked to appraise and critique a preliminary set of design guidelines identified from the literature and from prior studies [9–17,19,20]. The panel of experts were also asked to comment on each guideline as well as identify additional design guidelines that they thought could help address stigma via digital platforms. There were three rounds of feedback and additional items that were generated from participants' comments were integrated into the subsequent round. Details of each step are explained below.

### Ethics statement

This study was approved by the University of British Columbia Behavioral Research Ethics Board (Approval number = H21-00760). All participants provided written consent before participating in this study.

### Pre-Delphi stage

Consistent with prior Delphi studies, a candidate list of 28 items was generated through a literature review before the start of the Delphi study [21,22]. Using the concept of trauma-informed care as the guiding framework, we generated design guidelines that could help to alleviate stigma related to sexual health [23]. Trauma-informed care is a holistic approach to healthcare that seeks to offer an understanding of and a thoughtful response to individuals who have experienced an emotionally traumatic event(s), aimed at enhancing their resilience and self-efficacy [24]. Indeed, trauma-informed care is becoming an important area of interest in digital health research [25]. A trauma-informed care framework was considered useful for addressing stigma via digital platforms because of the intertwined relationship between people who experience stigma and trauma [26], and also because of how a trauma-informed framework has been successfully used in HIV stigma-reduction activities [27]. Given that some digital health technologies inadvertently perpetuate stigma [13,16], a different approach using a trauma-informed care framework was thought to have the potential to address stigma concerns among people who use digital health technologies. We specifically used the trauma-informed care framework proposed by Harris and Fallot [23]. Unlike traditional user-interface design guidelines that mostly concentrate on addressing technical problems [28], the use of this framework helped identify preliminary design guidelines from the literature review that were largely content-related and non-technical in nature. By adopting this framework, we were able to generate non-technical design solutions that cover a broad range of stigma management strategies [29–32].

### Participant recruitment

A list of 158 researchers in sexual health and stigma-related research across North America, Europe and Africa was purposefully compiled via publications, conference presentations, and

through the authors' professional networks. These researchers were subsequently recruited via email invitation. To be eligible, potential participants were required to have at least five years of experience in sexual health or stigma-related research, be fluent in English, and have a minimum of an undergraduate degree. The target sample size was 30 researchers, which is consistent with prior Delphi studies [22,33].

## Data collection

Data collection occurred in 3 rounds of Delphi and ended after the third round. Data from each round were analyzed before the next round began. In Round 1, the preliminary list of 28 design guidelines were circulated to the participants via an online survey platform (Qualtrics™). The participants rated the clarity and usefulness of each guideline in addressing sexual health-related stigma via digital platforms on a scale of 1–7, where 1 = strongly disagree, 2 = disagree, 3 = mildly disagree, 4 = neutral, 5 = mildly agree, 6 = agree, and 7 = strongly agree. Participants also justified their ratings by commenting on the usefulness and clarity of each item. At the end of the survey, participants were asked to provide additional ideas that they thought could help to address stigma. Items that achieved consensus in Round 1 were retained as part of the final set of design guidelines while those not achieving consensus were revised and fed back in Round 2. Additional design guidelines were also generated from participants comments and suggestions from Round 1. These additional guidelines were presented to participants in Round 2. In Round 2, participants appraised the clarity and usefulness of the revised and newly generated guidelines. Items that achieved consensus were retained as part of the final set of guidelines. Additional guidelines were also generated from Round 2 and the new items as well as items not meeting consensus were presented in Round 3. The same appraisal process occurred in Round 3. However, because this was the final round, no additional comments or suggestions were solicited from the participants. Instead, participants were asked to state their concluding remarks regarding the study. Consensus items were retained and those not achieving consensus were dropped. An item was considered as having been agreed upon if it obtained a mean value $\geq 5$ on the Likert scale (i.e., overall value greater than "mildly agree").

## Data analysis

At the end of each round, a content validity index (CVI) and interquartile range (IQR) were calculated for each guideline across the two dimensions of clarity and usefulness. Consensus in this study was defined as a CVI of $\geq 80\%$ and an IQR of $\leq 1$ on both clarity and usefulness dimensions. A CVI measures the percentage of people who agree with the clarity and usefulness of the guidelines in addressing stigma while an IQR measures the variation among participants' agreement or disagreement. That is, for an item to reach consensus, at least 80% of the panel should agree on its clarity and usefulness and the IQR for such an agreement should not be more than 1. An IQR was chosen because it is less affected by outliers and is also considered a better measure of dispersion for smaller sample sizes than a standard deviation [34].

In the second round, stability was calculated for both clarity and usefulness dimensions to determine the consistency of answers between Round 1 and Round 2. Stability refers to the consistency of answers between two successive rounds of a Delphi study [35]. It is calculated on items not meeting consensus to determine which items to drop. For Likert scale items, stability is often calculated by following a five-step process including: i) adding up the number of responses for each response option for each item in Rounds 1 and 2; ii) calculating the difference in the response option between Rounds 1 and 2 (for instance, if 5 people responded between "strongly disagree to neutral" for an item in Round 1 and 3 people choose the same

response in Round 2, then the absolute difference would be 2); iii) adding all the absolute differences across all items; iv) dividing the total sum by the number of rounds (2 rounds); and v) dividing the results from step 4 by the number of participants (14 participants between Round 1 and Round 2 in our study). The final absolute score could range from 0–1 where a higher score (typically above 0.2, or 20%) indicates wide variations in response options between the two rounds and a lower score (i.e., typically lower than 20%) suggests that responses for an item between two rounds will not change in another round. Stable items had responses between "strongly disagree" to "neutral" while unstable items had responses from "strongly disagree" to "strongly agree" on the Likert scale. The purpose of calculating stability was to establish which items were unlikely to change if sent to Round 3. Items were considered stable (i.e., if participants' responses ranged from strongly disagree to neutral) and were subsequently dropped if they achieved a stability value of ≤ 0.2 (or 20%) on both clarity and usefulness dimensions. Items with stability levels >0.2 (or 20%) in either 1 or both dimensions were considered unstable (i.e., if responses ranged from strongly disagree to strongly agree) and were revised for Round 3. Since Round 3 was the final round, stability was not calculated on items not achieving consensus.

## Results

### Characteristics of the Delphi panel

Of the 158 participants who were contacted, 22 (13.9%) expressed interest and met the eligibility criteria. Of these 15 participated in Round 1, 14 participated in Round 2, and 13 participated in Round 3. Of the 15 initial participants, 5 (33.3%) were from Canada, 5 (33.3%) were from Europe, 4 (26.6%) were from the United States, and 1 (6.6%) was from Rwanda. Ten (66.6%) participants self-identified as a woman, 4 (26.6%) identified as a man, and 1 (6.6%) did not report on gender identity. Thirteen (86.7%) were either assistant, associate, or full professors at research-intensive universities while 2 (13.3%) were working at public health agencies. The mean number of years of work experience in sexual health and stigma-related research was 21.6 years (SD = 10.26, Range = 6–46 years). 13 participants had a PhD, 1 had a Masters, and 1 had both an MD and a Masters.

### Delphi results

In Round 1, twelve (12) design guidelines achieved consensus. However, participants provided useful comments on 8 of these items that warranted further revisions (see tables below and S1 Table). Therefore, only 4 items were retained as part of the final set of design guidelines in Round 1. Of the 16 guidelines that did not meet consensus, 9 were revised, 2 were merged, and 5 were dropped (i.e., DG3, DG4, DG6, DG14, DG20, and DG24)–bringing the number of design guidelines for Round 2 to 26. In Round 2, 12 of the 26 guidelines achieved consensus and were added as part of the final guidelines. Two (2) items were dropped because they were either stable with "strongly disagree—neutral" scores on both clarity and usefulness dimensions while another 2 were dropped because the content was present in other newly generated guidelines. Ten items were thus revised, and two additional items were generated, making a total of 12 items for review in Round 3. In Round 3, only three of the 12 items achieved consensus. Since this was the final round, no further revisions were conducted on the guidelines that did not achieve consensus. At the end of Round 3, 19 design guidelines were found to have achieved consensus. Fig 1 shows a summary of the Delphi process, together with the number of items that achieved consensus, were revised, dropped, or retained for each round of the Delphi study. The figure also shows the number of respondents for each round of the

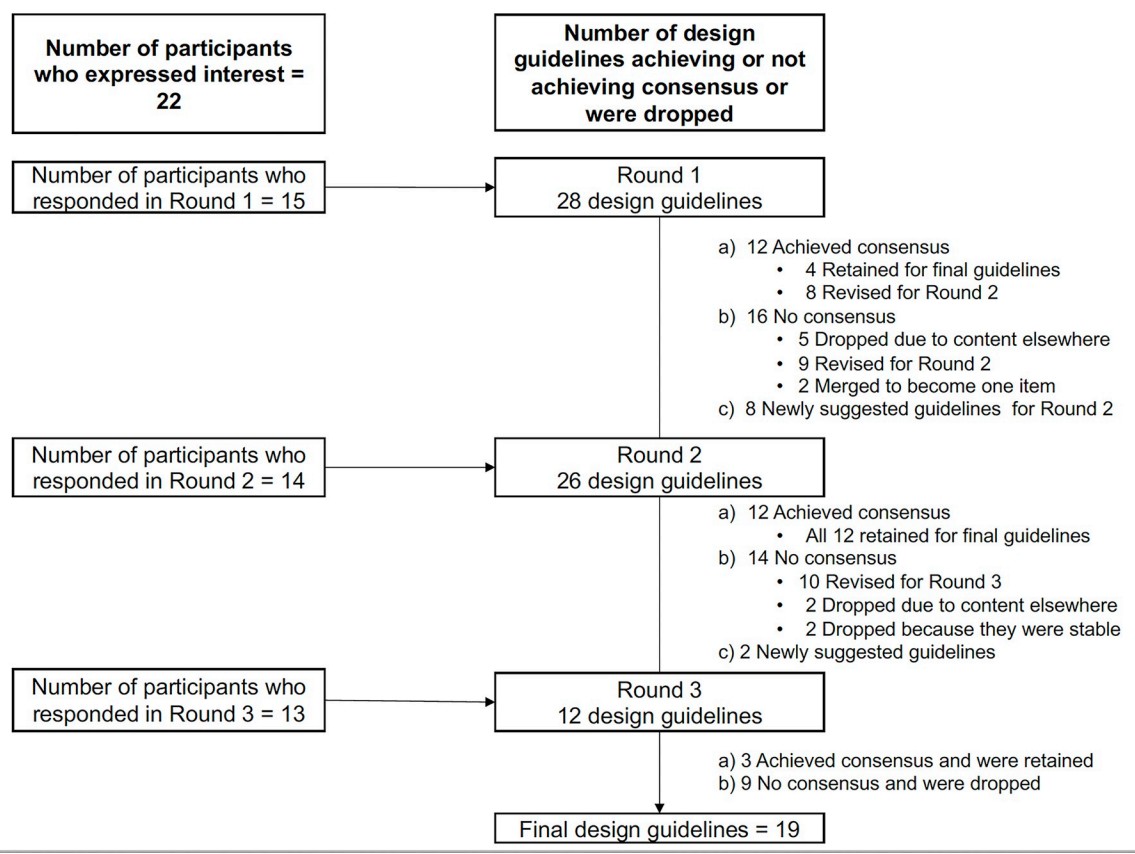

**Fig 1. Flowchart of the Delphi Technique Process.**

Delphi. Table 1 shows the detailed results for each round of the Delphi study for the original 28 guidelines.

## Revision of design guidelines not achieving consensus

The design guidelines that did not meet consensus were revised because they were perceived as either not clear, double-barrelled, or not useful/irrelevant to stigma reduction. Therefore, the revisions were mainly done to improve on items' clarity, usefulness, and relevance to stigma reduction via digital platforms. Some items were dropped or merged through the revision process because they were either thought to be redundant or could inadvertently aggravate rather than alleviate stigma. For instance, design guidelines that were meant to hide stigmatizing attributes on digital platforms (i.e., Ensure appropriate flexibility of web features [DG13]; Make logos, icons, and terminologies subtle [DG5]) were seen as likely to perpetuate or worsen stigma because such features might suppress people's feelings and their ability for expression. Similarly, design guidelines that centered on integrating links and hyperlinks on digital platforms [DG12] were considered as having an inherent security and privacy risk. Furthermore, design guidelines that centered on having website users narrate their experiences [DG21 & DG7.1] were considered inappropriate and not secure if there were no trained operational staff to moderate such platforms. See S1 Table for the original design guidelines and their corresponding revisions in Rounds 1 and 2. Additional guidelines were also generated by reframing participants' comments that reflected antistigma-informed approaches to digital technology design. As seen in Table 2 below, 8 additional guidelines were obtained from the

Table 1. Results for the newly generated design guidelines for Rounds 2 and 3.

| ID | Original Design Guideline | Round 1 | | Round 2 | | Round 3 | |
|---|---|---|---|---|---|---|---|
| | | Clarity | Utility | Clarity | Utility | Clarity | Utility |
| | | CVI (IQR) | CVI (IQR) | CVI (IQR) | CVI (IQR) | CVI (IQR) | CVI (IQR) |
| DG1 | Provide participants with the ability for anonymous engagement[†] | 100 (1) | 100 (1) | Retained in original form in R1 | | | |
| DG2 | Encrypt websites that collect personal information to prevent unauthorized access to personal data[†] | 100 (1) | 92.8 (1) | Retained in original form in R1 | | | |
| DG3 | Prevent errors by having features that warn users from taking actions that can lead to accidental disclosure of status | 35.7 (3) | 35.7 (4) | Dropped in R1 due to content elsewhere | | | |
| DG4 | Use non-sensational language and images on web platforms | 85.7 (2) | 92.8 (2) | Dropped in R2 due to content elsewhere | | | |
| DG5 | Logos, icons, and terminologies should be made subtle so that they do not immediately draw the users' or bystanders' attention to sexuality | 42.8 (3) | 35.7 (3) | 92 (1) | 69 (3) | 69.2 (4) | 61.5 (3) |
| DG6 | Reduce offensiveness of images by minimizing the display of explicit or profane sexual images on websites | 57.1 (4) | 50 94) | Dropped in R2 due to content elsewhere | | | |
| DG7 | Avoid conveying negative experiences associated with a condition and rather emphasize positive messaging | 71.4 (4) | 64.2 (4) | 92 (2) | 76 (3) | 84.6 (2) | 84.6 (2) |
| DG8 | Use humour in a very positive and sensitive manner to lessen the fear and seriousness associated with a condition | 57.1 (2) | 50 (1) | 76 (3) | 76 (3) | 92.3 (2) | 84.6 (2) |
| DG9 | Avoid using language that seeks to blame, or cast moral judgment on people living with or infected by a condition* | 100 (0) | 0/100 | 100 (1) | 100 (1) | Retained in R2 | |
| DG10 | Provide opportunities for selective and voluntary disclosure of status to provide emotional connection and support | 100 (1) | 100 (1) | 76 (3) | 84 (1) | Dropped in R3 due to stability | |
| DG11 | Provide interpersonal contacts for people to seek direct online counseling and other psychological support * | 85.7 (1) | 85.7 (1) | 92 (1) | 84 (1) | Retained in R2 | |
| DG12 | Include links to other credible sites and contact information for users to seek further information | 85.7 (1) | 85.7 (2) | 92 (2) | 76 (2) | 84.6 (1) | 53.8 (2) |
| DG13 | Ensure appropriate flexibility of web features to allow a user to hide certain content/images that they find offensive | 78.5 (2) | 57.1 (3) | 76 (3) | 69 (3) | 69.2 (4) | 84.6 (2) |
| DG14 | Ensure customization features to enable users to adapt aspects of the interface to suit their visual preference | 78.5 (3) | 71.4 (3) | Dropped in R2 due to content elsewhere | | | |
| DG15 | Expose people to a range of messaging that addresses different aspects of stigma* | 71.4 (4) | 8.5 (2) | 84 (2) | 69 (3) | 100 (1) | 92.3 (1) |
| DG16 | Include information that seeks to correct wrong perceptions and myths surrounding a particular condition to enable users to reject negative and inaccurate beliefs* | 92.8 (1) | 100 (0) | 100 (0) | 100 (1) | Retained in R2 | |
| DG17 | Educate users about stigma by providing factual and plain language information that normalizes and de-stigmatises sexual health-related conditions* | 92.8 (0) | 100 (0) | 100 (1) | 100 (1) | Retained in R2 | |
| DG18 | Use positive, credible, and diverse language and images* | 85.6 (1) | 100 (0) | 100 (1) | 100 (1) | Retained in R2 | |
| DG19 | Use images and pictures of people from a variety of backgrounds who have experienced the condition* | 92.8 (1) | 92.8 (0) | 100 (1) | 92 (1) | Retained in R2 | |
| DG20 | Ensure a match between the system and the users' culture | 64.2 (4) | 78.7 (2) | Dropped in R2 due to content elsewhere | | | |
| DG21 | Use personal stories of renowned and relatable personalities who have experienced the condition | 92.8 (2) | 85.7 (2) | 92 (1) | 76 (3) | Dropped in R3 due to stability | |
| DG22 | Provide information on the fundamental rights of people affected by or living with the condition. e.g. Rights of persons living with HIV/AIDS* | 92.8 (1) | 100 (2) | 100 (1) | 92 (1) | Retained in R2 | |
| DG23 | Use inclusive language such as "we, our, or us" and "you are not alone" * | 71.4 (3) | 64.3 (3) | 92 (1) | 100 (1) | Retained in R2 | |
| DG24 | Use symbols of optimism, encouragement, and hope | 71.4 (3) | 2/78.5 | Dropped in R2 due to content elsewhere | | | |
| DG25 | Provide mechanisms for creating online advocacy groups | 71.4 (3) | 64.3 (3) | 84 (1) | 76 (2) | 92.3 (1) | 76.9 (3) |
| DG26 | Include a video or written testimonial of real people to talk about stigma and their experiences with the condition* | 92.8 (1) | 85.7 (1) | 92 (1) | 100 (1) | Retained in R2 | |
| DG27 | Avoid othering and stereotyping people with a sexual health-related condition [†] | 100 (0) | 100 (0) | Retained in original form in R1 | | | |

(*Continued*)

**Table 1.** (Continued)

| | | Round 1 | | Round 2 | | Round 3 | |
|---|---|---|---|---|---|---|---|
| | | **Clarity** | **Utility** | **Clarity** | **Utility** | **Clarity** | **Utility** |
| DG28 | Interventions for addressing men's/women health or sexual health-related conditions should be developed and delivered in partnership with living with the condition † | 92.8 (1) | 92.8 (0) | Retained in original form in R1 | | | |

Note: DG = Design Guideline, CVI = Content Validity Index, IQR = Interquartile Range

† = Guidelines retained in their original format

* = Guidelines retained in revised format, R1 = Round 1, R2 = Round 2, R3 = Round 3.

revision process in Round 1, and 2 additional guidelines were obtained in Round 2. Because the additional guidelines were generated after the Delphi study had begun, they were appraised twice in Round 2 and only once in Round 3

## The final set of design guidelines

The study started with 28 items generated from the pre-Delphi stage and an additional 10 items generated from Rounds 1 and 2 (Total = 38). At the end of Round 3, 19 items representing 50% of the original (n = 28) and newly generated (n = 10) items were judged by the panel of experts as clear and useful in addressing stigma via digital platforms. Table 3 shows the final set of consensus guidelines mapped onto the Harris and Fallot five principles of trauma-informed care framework. Design guidelines on this table without a decimal point are items that achieved consensus following a single appraisal process (both original and revised items). Item IDs with ".1" suffix are guidelines that achieved consensus in Round 2 following Round 1 revisions, while item IDs with ".2" suffix indicate guidelines that achieved consensus in Round 3 following Round 2 revisions.

**Table 2. Results for the newly generated design guidelines for Rounds 2 and 3.**

| | | Round 2 | | Round 3 | |
|---|---|---|---|---|---|
| | | **Clarity** | **Utility** | **Clarity** | **Utility** |
| **ID** | **Newly Generated Design Guidelines in Round 2** | **CVI (IQR)** | **CVI (IQR)** | **CVI (IQR)** | **CVI (IQR)** |
| NG1 | Develop trust by providing options for different gender identities* | 92 (1) | 92 (1) | Retained in R2 | |
| NG2 | Ensure rigorous methods to know the target audience* | 84 (1) | 92 (1) | Retained in R2 | |
| NG3 | Have clear, factual and neutral information* | 92 (1) | 100 (1) | Retained in R2 | |
| NG4 | Provide factual content by including references to all information* | 84 (2) | 84 (2) | 100 (1) | 92.3 (1) |
| NG5 | Have a section for website users to narrate their stories | 76 (3) | 69 (3) | Revised for R3 | |
| NG6 | Have a function that allows people to leave the website immediately | 76 (1) | 69 (3) | 76.9 (2) | 84.6 (3) |
| NG7 | Create collaboration spaces (forums, chatrooms) on the website so that those with similar experiences can connect | 92 (1) | 76 (3) | Revised for R3 | |
| NG8 | Involve all those affected by the condition, not just people with the condition. e.g., partners of people with the condition | 76 (2) | 92 (2) | Revised for R3 | |
| | **Newly Generated Design Guidelines in Round 3** | | | | |
| NG9 | Consider involving all those affected by the condition, not just people with the condition (e.g., partners of people with the condition) * | | | 92 (1) | 100 (1) |
| NG10 | Ensure that careful consideration has been given when interactive features will be used. e.g., is a moderator required | | | 69.2 (2) | 84.6 (2) |

Note: NG = Newly Generated Guideline, CVI = Content Validity Index, IQR = Interquartile Range

* = Guidelines retained in a revised format, R1 = Round 1, R2 = Round 2, R3 = Round 3.

**Table 3. Final set of design guidelines mapped onto the 5 principles of trauma-informed care framework.**

| Category | ID | Destigmatising Design Guideline |
|---|---|---|
| **Emotional safety** | DG1 | Provide participants with the ability for anonymous engagement |
| | DG2 | Encrypt websites that collect personal information to prevent unauthorized access to personal data |
| | DG9.1 | Avoid using language that has a tone of blame or judgment of people living with the condition |
| **Choice** | DG11 | Provide contact information of counselors or other psychological supports |
| | DG15.2 | Include a range of evidence-based information that touches on different aspects of the condition |
| **Trustworthiness** | DG16.1 | Include information that corrects myths about a condition, to enable users to get accurate information |
| | DG17.1 | Provide factual and plain language information that normalizes and de-stigmatises sexual health-related conditions |
| | DG18.1 | Selection of language/images should be done in consultation with the community to ensure diversity |
| | DG19.1 | Use images of people from diverse ethnicity, age, and gender identity who have experienced the condition |
| | NG2 | Develop trust by providing options for different gender identities |
| | NG1 | Provide trustworthy content by having a reference list to factual information |
| | NG6 | Ensure rigorous methods to know the target audience |
| | NG8 | Have clear, factual and neutral information |
| **Empowerment** | DG22.1 | Consider including links to information on the fundamental rights of people affected by or living with the condition |
| | DG23.1 | Use inclusive language that is sensitive to the context of the condition e.g., partner instead of husband/wife, the person instead of woman/man |
| | DG26.1 | Include videos/testimonials that center on people's experiences with the condition, including stigma |
| | DG27 | Avoid othering and stereotyping people with a sexual health-related condition |
| **Collaboration** | DG28 | Interventions for addressing men's/women health or sexual health-related conditions should be developed and delivered in partnership with those living with the condition |
| | NG9 | Consider involving all those affected by the condition, not just people with the condition (e.g., partners of people with the condition) |

## Discussion

To the best of our knowledge, this study is the first to develop a set of technology design guidelines from a health-related domain by employing a trauma-informed care framework. Even though the usefulness of the trauma-informed care framework was not empirically tested in this study, the results align with previous studies that suggested that a trauma-informed care framework could help address stigma [27]. The findings of this study are expected to make a unique contribution to the body of literature on design guidelines as well as stigma management via digital platforms. Unlike traditional design guidelines that tend to focus on the technical solutions and the functionality/usability of digital technologies, this study demonstrates a shift from a predominantly technically focused approach to designing digital platforms to include emotional and content-related guidelines.

The development of content-related and largely emotional design guidelines in this study reinforces the need for and importance of incorporating emotional design requirements in developing digital health technologies [36]. Thus, the guidelines developed in this study are expected to help technology developers think beyond systems' usability and functionality features to include essential elements that can safeguard patients' emotional safety, including not contributing to stigma. The incorporation of content-related guidelines in designing digital

platforms is important because design guidelines that fall outside the usual technical guidelines are generally less obvious, more difficult to specify, and most often not considered by digital health developers [28]. By situating this study at the intersection of sexual health and digital health design, it can also be argued that the final design guidelines reflected human-centred values including human dignity, privacy, anonymity, autonomy and choice, sympathy, human relationships, and strength-based approaches, which are considered cardinal hallmarks of good healthcare [37].

Unlike prior stigma research that emphasizes hiding or avoiding strategies as appropriate mechanisms for alleviating stigma [29–32], this study suggested otherwise. Instead of hiding stigmatizing attributes as a way of addressing sexual health-related stigma, the design guidelines produced in this study rather seek to challenge, expose, and normalize ostensibly stigmatizing attributes via digital platforms. This open approach to addressing stigma reflects modern stigma management strategies that seek to shift the focus from the individual to include other people and larger societal factors that simultaneously operate to shape and perpetuate stigma [38–40]. A reason that was advanced against hiding or avoiding strategies is the fact that hiding stigmatizing content may rather lead people to accept public perceptions or accept the status quo regarding the stigmatizing attribute [30]. In other words, hiding or avoiding strategies may perpetuate internalized stigma by forcing people to accept the negative attributes associated with their condition. The findings of this study thus suggest that earlier stigma management strategies that center on hiding, avoiding, evading, or covering the stigmatizing attributes might have outlived their usefulness in addressing stigma in our current context.

The design guidelines produced in this study also reflected modern stigma management strategies that adopt education as a tool to increase peoples' knowledge about a specific condition. Education strategies involve interventions that aim to inform the public, community groups, and individuals by increasing their knowledge about a specific condition by providing facts that counterbalance the false assumptions upon which stigma is based [31]. Participants in this study had a positive consensus on two of the four principles that adopted education strategies to alleviate stigma. These principles include DG16.1 (i.e., Include information that corrects myths about the condition to enable users to get accurate information), and DG17.1 (i.e., Provide factual and plain language information that normalizes and de-stigmatizes sexual health-related conditions). Participants' agreement on the usefulness of education strategies in addressing stigma confirms the predominant use of educational messages in stigma reduction campaigns [31]. Participants not only agreed on the usefulness of these educational principles in addressing stigma but also agreed with principles that would make educational messages accurate, reliable, and non-judgmental. This reinforces earlier studies that suggest that educational messages that are accurate, reliable, credible, and non-judgmental are crucial for addressing stigma [41,42].

While education strategies were considered useful in addressing sexual health-related stigma, participants' comments suggested that the content of educational messages is very important for its effects on stigma and stigma reduction. Indeed, the effect of educational content on stigma reduction seems to be more pronounced in the information displayed on sexual health-related websites [13,43]. The manner in which educational messages are crafted and displayed on digital platforms could determine their success in alleviating stigma [9–16]. Even though the panel of experts argued against hiding or avoiding stigmatizing content, adopting the opposite approach by displaying content in a superficial and unfavourable manner could inadvertently perpetuate stigma in the end. To determine the right approach, careful consideration of the context in which the digital health technology will be developed would determine whether openly displaying sexual health-related content or being discrete with content would

help in addressing stigma. The effect of context-specific content display on stigma alleviation was not explored in this study. Further studies are needed to examine how, for example, the same web-based content impacts stigma in different cultural and linguistic settings.

The study also produced design guidelines that reflect the principles of inclusive design [44,45]. Unlike traditional inclusive design guidelines that are largely focused on designing products for older people and people with physical disabilities [44], this study extends the notion of inclusive design to include people with a range of ages, ethnicities, and gender identities [DG 19.1]. At a time when the digital divide is negatively impacting people from marginalized sex and gender identities, [46] consideration of the design guidelines produced in this study as belonging to the inclusive design sphere may help digital health teams to understand how to design digital health technologies that are inclusive of marginalized groups most often affected by stigma [46].

## Limitations

This study has some limitations that should be considered when interpreting or applying the findings. Unlike the preliminary design guidelines identified in the literature review, the 10 additional guidelines generated during the Delphi procedure did not benefit from a 3-round appraisal process. Also, the data collection method did not allow participants to discuss the topic amongst each other. Instead of recruiting only people with research expertise on the topic, the Delphi process could have benefited from a review by experts in the field of sexual health practice such as nurses, physicians, and social workers. Further studies should convene a focus group among researchers and field experts as well as people who use digital health interventions to further understand how the design guidelines can be applied. Finally, out of 158 participants contacted, only 22 agreed to participate, and 13 participated in all three rounds. It is unknown if the demographics of participants who agreed to participate or those who participated were truly representative of the 158 people who were originally contacted.

## Conclusion

The findings of the Delphi study extended the knowledge on design guidelines from a predominant focus on technical and usability considerations of digital health technologies to include emotional and content-related guidelines. Design guidelines that transcend the usual functionalist considerations are often not considered by development teams yet considered critical for ensuring emotional comfort of the end-users. Unlike the earlier stigma management strategies, the design guidelines also reflect a modern approach to managing stigma by suggesting that the stigmatizing attribute be made public via digital platforms. This public approach has the potential to ultimately make stigma more of a community or societal concern rather than an individual attribute. For the design guidelines to be successful in addressing stigma, project managers or organizations that wish to embark of digital health projects for sensitive health topics like sexual health should consider making this reference guide an essential part of the project contract when engaging with the development team. This way, digital health developers can ensure that the final product will strive to prevent stigma.

## Supporting information

**S1 Table. Design guideline revisions at Rounds 1 and 2.**
(DOCX)

## Author Contributions

**Conceptualization:** Abdul-Fatawu Abdulai.

**Data curation:** Abdul-Fatawu Abdulai.

**Formal analysis:** Abdul-Fatawu Abdulai.

**Funding acquisition:** Abdul-Fatawu Abdulai, A. Fuchsia Howard, Paul J. Yong, Leanne M. Currie.

**Investigation:** Abdul-Fatawu Abdulai.

**Methodology:** Abdul-Fatawu Abdulai, A. Fuchsia Howard, Leanne M. Currie.

**Project administration:** Abdul-Fatawu Abdulai, Paul J. Yong.

**Supervision:** A. Fuchsia Howard, Paul J. Yong, Leanne M. Currie.

**Validation:** A. Fuchsia Howard, Paul J. Yong, Leanne M. Currie.

**Writing – original draft:** Abdul-Fatawu Abdulai.

**Writing – review & editing:** Abdul-Fatawu Abdulai, A. Fuchsia Howard, Paul J. Yong, Leanne M. Currie.

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
