## [Decision Letter · Decision Letter 0]

9 May 2023

PDIG-D-23-00061

Defining Destigmatizing Design Guidelines for Use in Sexual Health-Related Digital Technologies: A Delphi Study

PLOS Digital Health

Dear Dr. Abdulai,

Thank you for submitting your manuscript to PLOS Digital Health. After careful consideration, we feel that it has merit but does not fully meet PLOS Digital Health's publication criteria as it currently stands. Therefore, we invite you to submit a revised version of the manuscript that addresses the points raised during the review process.

Please submit your revised manuscript within 30 days Jun 08 2023 11:59PM. If you will need more time than this to complete your revisions, please reply to this message or contact the journal office at digitalhealth@plos.org. Please include the following items when submitting your revised manuscript:

We look forward to receiving your revised manuscript.

Kind regards,

Shlomo Berkovsky

Section Editor

PLOS Digital Health

Journal Requirements:

2. Please provide separate figure files in .tif or .eps format only and remove any figures embedded in your manuscript file. Please also ensure that all files are under our size limit of 10MB.

Additional Editor Comments (if provided):

Many thanks for your submission. After reviewing the comments of the reviewers, I recommend Acceptance with minor revisions. In order to ensure the highest quality of publication, we kindly request that you address the comments and suggestions provided by our reviewers. These revisions are essential in maintaining the standards of our journal and will help enhance the clarity and impact of your work. Please carefully consider and incorporate these comments into your revised manuscript.

Reviewers' comments:

Reviewer's Responses to Questions

**Comments to the Author**

1. Does this manuscript meet PLOS Digital Health’s publication criteria? Is the manuscript technically sound, and do the data support the conclusions? The manuscript must describe methodologically and ethically rigorous research with conclusions that are appropriately drawn based on the data presented.

Reviewer #1: Yes

Reviewer #2: Yes

2. Has the statistical analysis been performed appropriately and rigorously?

Reviewer #1: Yes

Reviewer #2: Yes

3. Have the authors made all data underlying the findings in their manuscript fully available (please refer to the Data Availability Statement at the start of the manuscript PDF file)?

Reviewer #1: Yes

Reviewer #2: No

4. Is the manuscript presented in an intelligible fashion and written in standard English?

Reviewer #1: Yes

Reviewer #2: Yes

5. Review Comments to the Author

Reviewer #1: Thank you for the opportunity to review your manuscript. I believe the manuscript is an important contribution to the state of science for development of de-stigmatizing sexual health-related digital technologies. The manuscript provides a clear description of the study methodology and results from the Delphi study to define the design guidelines. This is an important issue for IT professionals and researchers who develop sexual-health related technologies. From my perspective, there are no substantive or content specific changes needed. I believe there evidence the study was conducted ethically and I have no concerns about the ethical considerations described in the manuscript. 

My suggested edits are related to a few minor editorial concerns that are outlined below. Page numbers are based on the full pdf including the introductory pages attached by the journal submission system. 

1. Please confirm whether "ethics committee" should be capitalized as part of the name of the UBC Research Ethics Board. 

2. On page 11, I think the word "indicate" should be plural when you describe the scale, i.e. "...where 7 indicates..."

3. Pages 20-21, in Table 3, consider shifting the first column so the framework category titles are not hyphenated. 

4. On page 22, please format the references for Friedman et al., Hood & Friedman, and Rocha-Jimenez are formatted consistent with the other references in the manuscript. 

5. On page 23, clarify whether "NDP35" is correct. 

6. On page 23, confirm whether you need the word "based" at the end of the sentence ending with "...the predominant use of educational messages in stigma reduction campaigns based [30]." 

7. On page 24 in the the limitations section, I think you can remove the word "adopt" before the phrase "... a focus group among experts..."

Reviewer #2: Dear authors, 

Thank you for submitting the study: „Defining Destigmatizing Design Guidelines for Use in Sexual Health-Related Digital Technologies: A Delphi Study“.

The study describes the development of 19 guidelines for the destigmatizing design of digital sexual health solutions using the Delphi method. In three rounds, 13 researchers were consulted. The topic has great practical relevance for development teams and the method is quite suitable for developing those guidelines. The manuscript is written in a concise and understandable manner. Attached you will find comments on minor issues that should be addressed prior to publication. 

Kind regards

6. PLOS authors have the option to publish the peer review history of their article (what does this mean?). If published, this will include your full peer review and any attached files.

**Do you want your identity to be public for this peer review?** For information about this choice, including consent withdrawal, please see our Privacy Policy.

Reviewer #1: Yes: J. Craig Phillips, LLM, PhD, RN, ACRN, FAAN, FCAN

Reviewer #2: Yes: Marlene Muehlmann

---

## [Editor Report · Decision Letter 1]

28 May 2023

Defining Destigmatizing Design Guidelines for Use in Sexual Health-Related Digital Technologies: A Delphi Study

PDIG-D-23-00061R1

Dear Dr. Abdulai,

We are pleased to inform you that your manuscript 'Defining Destigmatizing Design Guidelines for Use in Sexual Health-Related Digital Technologies: A Delphi Study' has been provisionally accepted for publication in PLOS Digital Health.

Best regards,

Baki Kocaballi

Academic Editor

PLOS Digital Health

Many thanks for submitting your revised paper. The revisions have addressed the reviewers' comments in a reasonable way. I would recommend accepting this paper.